# Conditions to Control Furan Ring Opening during Furfuryl Alcohol Polymerization

**DOI:** 10.3390/molecules27103212

**Published:** 2022-05-17

**Authors:** Lucie Quinquet, Pierre Delliere, Nathanael Guigo

**Affiliations:** Institut de Chimie de Nice, Université Côte d’Azur, CNRS, UMR 7272, 06108 Nice, France; lucie.quinquet@univ-cotedazur.fr (L.Q.); pierre.delliere@univ-cotedazur.fr (P.D.)

**Keywords:** biobased poly(furfuryl alcohol), ring-opening, degree of open structures

## Abstract

The chemistry of biomass-derived furans is particularly sensitive to ring openings. These side reactions occur during furfuryl alcohol polymerization. In this work, the furan ring-opening was controlled by changing polymerization conditions, such as varying the type of acidic initiator or the water content. The degree of open structures (DOS) was determined by quantifying the formed carbonyl species by means of quantitative ^19^F NMR and potentiometric titration. The progress of polymerization and ring opening were monitored by DSC and FT-IR spectroscopy. The presence of additional water is more determining on ring opening than the nature of the acidic initiator. Qualitative structural assessment by means of ^13^C NMR and FT-IR shows that, depending on the employed conditions, poly(furfuryl alcohol) samples can be classified in two groups. Indeed, either more ester or more ketone side groups are formed as a result of side ring opening reactions. The absence of additional water during FA polymerization preferentially leads to opened structures in the PFA bearing more ester moieties.

## 1. Introduction

In recent years, research on biobased alternatives to fossil-based polymers has been developed as part of a general concern for sustainability [1]. Some, such as poly(lactic acid), can be obtained from starch [2]. However, arable lands are necessary in order to produce said starch, thus competing with food crops. Extensive studies have been conducted to substitute petroleum-based chemicals while avoiding such competition. Lignocellulosic biomass is an interesting resource since it can be obtained through agricultural wastes. It is constituted of lignin, cellulose, and hemi-cellulose [3] that can be processed into phenols, hexoses, and pentoses before being further converted into platform molecules [4]. Lignin may yield phenolic precursors that can partly substitute petroleum-based phenol for the preparation of phenol-formaldehyde resins [5,6]. The hexoses can be converted into hydroxymethylfurfural, a platform chemical that can be oxidized into 2,5-furandicarboxylic acid. This latter can be polymerized into poly(ethylene 2,5-furandicarboxylate), a serious candidate as a replacement for poly(ethylene terephthalate) [7]. Finally, pentoses can be dehydrated into furfural, another key platform molecule [8,9]. Furfural is most widely used for the production of furfuryl alcohol (FA) by catalytic hydrogenation [4]. FA can also be directly produced from xylose using one-pot systems [10]. The former can be polymerized into poly(furfuryl alcohol) (PFA) through acid catalyzed polycondensation (Figure 1). This complex polymerization has been extensively studied [11,12,13,14,15,16,17,18,19]. From these studies, two main steps were identified in the polymerization. The first one consists in oligomer formation by FA polycondensation and the second is a branching via Diels–Alder cycloaddition, leading to a high crosslinking density. The tightness of the networks induces the strong brittle behavior of the material. This can limit its uses as a thermoset resin for structural applications when resilience and/or toughness are demanded. As a consequence, PFA is rarely used on its own and is mostly used in composites with reinforcement, such as flax fibers [20] or carbon fibers [21]. Currently, PFA is mostly used to bind sand particles together into foundry molds [22].

The ring opening reaction of furfuryl alcohol yields levulinic acid (LA) or its related esters. The conversion of FA through this pathway has recently been a subject of interest since LA is an interesting platform molecule [23,24,25,26]. However, few studies have focused on the ring-opening reactions within PFA. Falco et al. [27] studied the effect of the furan ring opening on the structure and properties of PFA. Indeed, the furan ring can cleave into carbonyl containing moieties within the polymer [18,19,28]. They showed that the polymerization of FA with protic polar solvents (water, isopropanol) leads to more open structures, which have been associated to a lower cross-link density and lower glass transition temperature than the neat system.

Since carbonyl containing species such as ketones are formed during the furan ring opening, it is interesting to quantitively follow the amount of carbonyls in the reacting system. Several techniques have been developed to quantify the ketones in a product, such as the Faix method [29] modified by Black [30] via potentiometric titration and quantitative ^19^F NMR [31]. These methods have been recently adapted to carbonyls formed within PFA in a study by Delliere et al. They investigated how the degree of opened structures progresses with the course of polymerization of FA [32] and proposed a new quantification metric for these opened structures, namely the ’degree of open structures (DOS)’. It is defined as the moles of opened structures from furan opening per moles of furan in the sample. It is assumed that each open structure corresponds to two ketones which were quantified using carbonyl quantification as formerly explained. Following the example depicted in Figure 1, there is theoretically 14 furfuryl units but two of them have been opened into carbonyl containing species and two merged through a Diels Alder reaction. The resulting DOS of 0.14 was calculated as follows:(1)$$\mathrm{DOS}=\frac{2\;open\;structures}{14\;furfuryl\;units}=0.14$$

The DOS will thus range from 0 to 1 assuming that one furfuryl unit will give a maximum of two ketonic species.

To our knowledge, no studies were devoted to the effects of the polymerization reaction conditions on the ring opening reaction within PFA.

Accordingly, in this work, the conditions of the furan ring opening during polymerization were studied.

First, the role of the acidic initiator on the furan ring opening was studied by comparing Brönsted initiators, such as citric acid, oxalic acid, and acetic acid, with Lewis initiators, such as boron trifluoride, iodine, alumina, or montmorillonite K10 (MMT-K10). Most have already been used in the past [33,34,35]. A Brönsted superacid, such as trifluoromethanesulfonic acid, as well as initiators combining the two acidities such as cation exchanged montmorillonite with alkyl-ammonium, were also employed. Indeed, cation exchanged montmorillonite (Org-MMT) displays a dual acidity which, as highlighted by Zavaglia et al. [34], leads to an acceleration of the FA polymerization rate. Therefore, the dual acidity of Org-MMT might arguably affect the ring opening occurring during polymerization. The effect of additional water on the ring opening was investigated as well.

In a final part, the nature of the carbonyl groups was investigated using PFAs from the aforementioned experiments as well as PFAs from syntheses using isopropanol as a solvent. They were then compared by displaying the DOS_Titri_ as a function of the C=O area of the corresponding FTIR spectra.

## 2. Results and Discussion

The main purpose of this work was to study the furan ring opening, focusing on certain aspects, such as the effects of the acidic initiator, the presence or absence of additional water, and its ratio on the ring opening.

### 2.1. Screening of the Different Initiators

As stated previously, different initiators were investigated. For the Brönsted type, citric acid and oxalic acid were chosen as they have already been used for preparing PFA, as well as acetic acid for its industrial accessibility [35].

A Brönsted superacid, namely trifluoromethanesulfonic acid (TfOH), was also investigated.

As for the Lewis type, boron trifluoride in methanol solution was chosen for being in liquid state. Alumina, iodine, and MMT-K10 have been used in the past [11,33,36].

Montmorillonite (MMT) belongs to the family of smectites clays which are considered as selective, safe, efficient, and eco-friendly initiators. It is a layered aluminosilicate in which the silica and alumina form a sheet like structure. The hydrophilic nature of the clays is explained by the presence of alkali charge compensating counterions within the interlayer spaces. These counterions can be exchanged with organic or metallic cations. MMT exhibits both Brönsted and Lewis acid sites. The Brönsted acidity is mainly due to dissociation of the intercalated water molecules coordinated to cations. The Lewis acid sites are located on alumina sheets. Accordingly, MMT is of great interest in the case of FA polymerization. Indeed, its dual acidic character could catalyze both condensation and addition during the FA polymerization [34].

Therefore, Sodium MMT (MMT-K10) and an organically modified MMT obtained by exchanging the counterions with octadecyl ammonium cations (Org-MMT) were used in this study to separately study the effect of the Lewis acid site with MMT-K10 and the effect of the dual acidity with Org-MMT.

The results of the investigations on the role of the initiator are shown in Figure 1 for the titration results and Appendix A for the NMR results. All results are indexed in the Appendix A. The DOS varies significantly but there is no clear demarcation between the different types of acidity.

The results plotted in Figure 1 suggest that the nature of acidic initiator (i.e., Lewis vs. Brönsted) does not particularly affect the DOS. Indeed, at a similar reaction progress, Lewis and Brönsted acids do not notably stand out from one another, and neither do the combined acidities.

The results from the ^19^F NMR method are available in Appendix A. A point worth noting is the undeniable difference between the titration and NMR results. They follow the same trend. However, the DOS_Titri_ is always superior to the DOS_NMR_. As explained in a previous work [32], this gap may be explained by the difference in size, and therefore steric hindrance, of the derivatization agents used in both cases to react with the ketones. The potentiometric titration uses hydroxylamine, unlike the NMR titration which uses much larger hydrazines, such as 4-(trifluoromethyl)phenylhydrazine. The differences in the DOS of these two techniques demonstrate the interest in using both of them, giving in the end an overall range of the DOS.

These investigations showed that the DOS progressively increases with the conversion degree. However, the type of acidity does not particularly influence the DOS.

Another aspect worthy of examination was the role of added water on the DOS.

### 2.2. The Role of Additional Water

In order to determine whether the presence of additional water and its ratio affect the DOS, two approaches were implemented.

#### 2.2.1. DOS Comparison with and without Additional Water

First, the effect of additional water was studied by conducting several syntheses in which an array of initiators was tested with and without additional water in a same amount in the system, i.e., the initial water amount, which will come in addition to the water generated during the polycondensation. The FA/water ratio was fixed to 50/50 *w*/*w*. The results of the first approach are indexed in Appendix A. For each initiator, the water analogue has a higher DOS.

The results plotted in Figure 2 suggest that the presence of water does influence the DOS by increasing it. The ^19^F NMR method in Appendix A presents the same trend. Indeed, at a similar conversion degree, the DOS is higher when the reaction was performed in the presence of additional water.

As stated, the results of this investigation show that the added water plays a role in the furan ring opening in agreement with both Falco et al. and Delliere et al. [27,32]. In addition to this seminal investigation, it is clearly highlighted here that the increase in DOS is not related to the acidic initiator. In other words, all the different FA/initiator systems show a DOS increase with additional water.

#### 2.2.2. The Role of the Additional Water Ratio

Secondly, several polymerizations were conducted by selecting only one initiator, namely maleic anhydride (MA), and changing the FA/additional water ratio, ranging from neat (100/0) to 40/60.

The results of the FA/additional water ratio investigations are indicated in Appendix A. Figure 3 exhibits the DOS_Titri_ as a function of the conversion degree for PFAs synthetized with more or less additional water. The ^19^F NMR results are depicted in Appendix A.

Figure 3 and Appendix A highlight that the DOS is continuously increasing with the conversion degree, independently of the FA/water ratio. The neat PFA shows the lowest DOS overall from the beginning to the near end of the polymerization, i.e., α ≈ 0.9. Regarding the water ratios, systems with less water (40/60 and 10/90) have slightly higher DOS than the 50/50 and 70/30 ratios at α < 0.80. Nonetheless, at higher conversion degrees, i.e., α ≈ 0.9, the gap shrinks around a DOS of 0.15. This is most likely due to the insolubility of PFA in water. Indeed, FA and water are miscible, whereas above α ≈ 0.20 a demixion occurs, as depicted in Figure 3. Thus, small, open, and water-soluble oligomers might not be taken into account in the DOS at low conversion degrees.

Then, when the polymerization progresses, they would be integrated in the PFA phase, leading to a final DOS of 0.15.

Consequently, these investigations allowed to determine that adding water plays a role in the opening of the furan ring, but the ratio of added water does not impact the DOS that is reached at the end of polymerization.

#### 2.2.3. Carbonyl Groups in PFA

A variety of PFAs were synthetized using the aforementioned catalysts with and without additional water. Secondly, syntheses using isopropanol (50/50 to FA) were performed. The DOS_Titri_ was measured as well as the FTIR C=O area, which is obtained by normalizing the FTIR spectra followed by an integration between 1840 and 1650 cm^−1^. In a study of C=O content in lignins, Faix et al. used this concept of FTIR C=O area to quickly determine the C=O content of lignins [29]. They obtained excellent linear correlation between the C=O content and the FTIR C=O area (R^2^ > 0.98). The goal of this study was to check if determining the DOS of PFAs can be somehow correlated with the FTIR C=O area.

The FTIR spectra of neat PFA and 50/50 water PFA catalyzed by 2%w MA at conversion degrees α ≈ 0.5 and α ≈ 0.8 are shown in Figure 4A. Both samples exhibit a C=O band at 1710 cm^−1^. This band has been attributed to ketonic moieties. However, the neat PFA exhibits a shoulder around 1780 cm^−1^, which has been attributed to ester moieties [19,27]. Figure 4B presents the ^13^C NMR of the same samples. The peaks corresponding to ketones appear between 195 and 210 ppm and, as shown in Figure 4B, the sample PFA 50/50 exhibits more peaks in this region compared to the neat PFA. On the other hand, the resonance attributed to esters are located between 160 and 178 ppm. The neat PFA shows more peaks attributed to esters compared to PFA 50/50. In PFA 50/50, where α ≈ 0.8, the 174 ppm peak is attributed to the ester band from the levulinic acid. Indeed, low levels of acid can be detected in such PFAs [32].

Figure 5 shows the DOS_Titri_ as function of the FTIR C=O area for all the investigated PFAs. A separation between PFAs prepared with additional water, the ones prepared in neat conditions, and the ones prepared with additional IPA is portrayed in Figure 5. It is worth mentioning that both ketones and aldehydes functions are quantified by the oximation method, but this is not the case for the ester function. However, it is rather difficult to discriminate the ketones/aldehydes from the ester functions when integrating the more or less broad carbonyl peak in FTIR. Accordingly, if only ketones or aldehydes are present in the systems, the C=O content should increase linearly with the FTIR C=O area. However, if other functional groups such as esters are taken into account in the C=O area, the DOS vs. the C=O area will not be linear.

Moreover, in their work, Wewerka et al. used alumina as a catalyst and identified lactones in the system [11]. In Figure 5, the data point of alumina catalyzed PFA (0.094; 30.4) is indeed in the ester bearing group.

Regarding the PFAs prepared with 50/50 IPA, they also belong to the ester bearing groups in Figure 5 as observed by Falco et al. [27]. Yet, they have a higher DOS than the neat PFA. This suggests an opening of furan rings through the formation of compounds, such as levulinates or lactones, thus increasing both the DOS and FTIR C=O area.

Finally, the ^13^C NMR spectra of PFA 50/50 in Figure 4B exhibits a significant number of carbonyl peaks from 194 to 210 ppm. Most of them are centered around 208 ppm, which matches with 1,4-diketones. Similar peaks can be found in the neat PFA. However, at α ≈ 0.80, new peaks rise in the 194–205 ppm area, in PFA 50/50 only. This confirms the existence of types of carbonyl containing species other than 1,4-diketones in this system.

These results suggest that additional water either prevents the formation of esters during FA polymerization or hydrolyses them or both.

## 3. Materials and Methods

### 3.1. Materials

Furfuryl alcohol (FA) (96%), 4-(trifluoromethyl)phenylhydrazine (96%), Hydroxylamine hydrochloride (99%), Maleic anhydride (99%), Citric acid (99.5%), Oxalic acid (99%), Acetic acid (99.5%), Isopropanol (98%)1,4-Bis(trifluoromethyl)benzene (98%), Triethanolamine (99%), trifluoromethanesulfonic acid (98%), Boron trifluoride-methanol solution (14%), and MMT K10 (Montmorillonite K10) were supplied by Sigma-Aldrich. Organically modified montmorillonite (Org-MMT), obtained by the exchange of MMT counterions with octadecyl ammonium cations, was supplied from Nordmann, Rassmann GmbH (Nanomer I30E). DMSO-d6 (99.80%) was supplied Euristop. Chromium(III) acetylacetonate (97%) was supplied by Alfa aesar. Iodine resublimed was supplied by Carlo Erba. Aluminoxid 90 neutral was supplied by Carl Roth. Ethanol (96 v%) was supplied by VWR. Water was distilled using SI Analytics distillation unit purchased from Thermo Fischer Scientific, France (conductivity ~1 µS/cm). All chemicals were used as received.

### 3.2. Methods

#### 3.2.1. Poly(furfuryl alcohol)’s Synthesis

Each synthesis was carried out in the following way: For syntheses without additional water, 25 g of FA were first added in a 100 mL round bottom flask. The appropriate amount of catalyst (Table 1) was then added under vigorous stirring. After complete solubilization of the catalyst, the mixture was then heated at 80 °C under reflux using a refrigerating tube and under vigorous stirring for an hour, samples were taken regularly and inspected via IR spectrometry.

After one hour, the refrigerating tube was taken off and the mixture was raised to 110 °C in order to remove the remaining water. The reaction was stopped when the mixture became a dark brown resinous rubber, still elastic, not hard. For syntheses with additional water, the appropriate amount of water (Table 1) was first added in a round bottom flask. The process is then identical to the one previously described. Depending on the initiator, the time of reaction varied. However, it never exceeded eight hours.

#### 3.2.2. Liquid State Nuclear Magnetic Resonance (NMR)

NMR spectra were recorded on a Bruker AVANCE III (400 MHz for ^19^F, 125.77 MHz for ^13^C) using d_6_-DMSO. ^19^F NMR spectra were carried out using a standard fluorine 90° pulse program, a relaxation delay of 3 s, and 256 scans acquired between −35 and −85 ppm. The processing of spectra was conducted with MestReNova software, and baseline (Breinstein polynomial order 3) and phase correction were applied. Finally, the spectra were referenced to internal standard at −62.15 ppm. The ^13^C NMR spectra were recorded with a 2 s relaxation delay and 6000 scans between −27.4 and 266.3 ppm. A high number of scans was used to ensure good signal/noise ratio in the carbonyl area.

#### 3.2.3. Differential Scanning Calorimetry (DSC)

The DSC measurements were performed on a DSC1 from Mettler Toledo equipped with a FRS5 sensor (with 56 thermocouples Au-Au/Pd). The temperature and enthalpy calibrations were performed using indium and zinc standards and the data were analyzed with STARe software. The samples weighted from 5 mg to 10 mg. Volatilization of the polycondensation byproduct occurs during the curing of the different mixtures, which is why the samples were placed into 30 μL high-pressure gold-plated crucibles. According to previous work, the conversion degree (α) was calculated using the following Equation (2) [34,37,38]:(2)$$\alpha =\frac{Q_{\mathrm{Max}}-Q_{\mathrm{residual}}}{Q_{\mathrm{Max}}}$$

Here, $Q_{\mathrm{Max}}$ is the total heat released during the polymerization. $Q_{\mathrm{residual}}$ corresponds to the heat generated when completing the polymerization of oligomers.

#### 3.2.4. Fourier-Transform Infrared Spectroscopy (FTIR)

The FTIR (ATR) measurements were recorded on a Thermo scientific Nicolet iS20 spectrometer from 400 to 4000 cm^−1^ with 64 scans and a resolution of 4.0 cm^−1^. A spectrum of air was used as background. The acquired spectra were analyzed with the OMNIC 9 software. First, a baseline was applied to the spectra. Then, they were normalized by using the C-O band at 1000 cm^−1^ as a reference band (absorbance = 1). The normalization is required to avoid variations caused by the refraction indexes. Indeed, it is the most intense band within both PFA and furfuryl alcohol [16,39]. The FTIR C=O area was set to (1650–1840 cm^−1^), according to literature [40].

#### 3.2.5. Carbonyl Quantification Methods and DOS Determination

In order to determine the DOS of the PFA resins, two carbonyl quantification methods adapted to furanic macromolecular systems were used as developed by Delliere et al. [32].

The first method, based on the work of Faix et al., uses the oximation and was conducted as follows [29,30]: About 100 mg of PFA were accurately weighted in a vial, followed by the addition of 2 mL of DMSO. The vials were then heated for 5 min at 80 °C in order to quickly dissolve the PFA. This step was followed by the addition of 2 mL of a 0.44 M hydroxylamine hydrochloride-EtOH solution (80 v%) and 2 mL of a 0.52 M triethanolamine-EtOH solution (96 v%). Moreover, blanks were performed, and the samples were triplicated. The last step consisted in heating the samples for 24 h at 80 °C in order to reach total oximation. The samples were then allowed to cool-down before titration with a 0.1 N HCl solution.

Consequently, the carbonyl content was calculated using the following equation:(3)$$CO\;content\;\left(\mathrm{mmol}/g\right)=\frac{\left(V_{0}-V\right)}{m_{\mathrm{PFA}}}*\left[\mathrm{HCl}\right]$$
where $V_{0}$ is the endpoint of the blank, V is the endpoint of the sample, and $m_{\mathrm{PFA}}$ stands for the mass of PFA in the sample.

The other method employed for carbonyl quantification was NMR based on the work of Constant et al. [31]. Quantifications were conducted as follows:

About 100 mg of PFA, 140 mg of 4-(trifluoromethyl)phenylhydrazine (CF_3_FNHNH_2_) and 20 mg of 1,4-bis(trifluoromethyl)benzene were accurately weighted in a 1.5 mL vial. Thereafter, 400 µL of *d_6_*-DMSO were added and the vial was shaken for 10 min before 5 s of centrifugation. Then, the content of the vials was transferred into NMR tubes and underwent heating for 24 h at 40 °C. Finally, 100 µL of a relaxing agent solution were added and homogenized just before analysis (28 mg/mL of chromium(III) acetylacetonate in *d_6_*-DMSO). The carbonyl content can be calculated through the following equation using the integrated areas from the acquired spectra:(4)$$CO\;content\;\left(\mathrm{mmol}/g\right)=\frac{\left[A_{c}-\left(\frac{A_{H{}^{\circ}}*m_{H}}{m_{H{}^{\circ}}}\right)\right]*m_{IS}}{A_{IS}*0.5*m_{PFA}*M_{IS}}*10^{3}$$
where $A_{c}$ stands for the hydrazones’ areas. $m_{IS}$, $M_{IS}$ and $A_{IS}$ correspond respectively to the mass, molecular mass, and the integrated area of the standard. Since an impurity appears to be present in the commercial CF_3_FNHNH_2_ its amount was determined in every batch. This results in a correction factor where $A_{H{}^{\circ}}$ is the integrated area of the impurity in the reference and $m_{H{}^{\circ}}$ is the mass of hydrazine within it. $m_{H}$ is the mass of hydrazine within the sample. Finally, the area of the standard was multiplied by 0.5 as it bears two CF_3_ groups.

At last, as described in a previous work, PFA resins can be characterized by their degree of open structures [32]. Indeed, as described in earlier publications, carbonyls in PFAs emerge from the opening of a furan ring into 1,4-dicarbonyls [18,19]. Thus, the DOS can be defined as follows:(5)$$\mathrm{Degree}\;\mathrm{of}\;\mathrm{open}\;\mathrm{structures}\;\left(\mathrm{DOS}\right)=\frac{1}{2}.\frac{N_{C=O}}{N_{furanic\;}}$$
where $N_{C=O}$ is the moles of carbonyls and $N_{furanic\;}$ the moles of furan units, i.e., furan-CH_2_–. This can be converted into Equation (6) for the carbonyl content to appear in the equation.
(6)$$\mathrm{DOS}=\frac{1}{2}\cdot 1000\cdot \;CO\;content\cdot \;M_{\left(furfuryl\;unit\right)\;}$$

The DOS obtained from the NMR method will be called DOS_NMR_ while the one resulting from oximation will be DOS_Titri_.

## 4. Conclusions & Prospects

In the present study, the conditions impacting the furan ring opening occurring during FA polymerization were investigated. Accordingly, the degree of open structures (DOS) was used as an indicator. Different acidic catalysts, spanning from Brönsted to Lewis acids, were employed to initiate FA polymerization. However, the results point towards the fact that no specific type of acidity particularly promotes or reduces the DOS.

Then, the influence of the additional water content was examined and different initial FA/water ratios ranging from 100/0 to 10/90 were employed. When FA polymerization is not too advanced (i.e., α < 0.8), the DOS is quite influenced by the water ratio. Logically, a higher water content will lead to a higher DOS. However, for α > 0.8, the DOS values are merging for all the FA/water ratio to an ultimate value of about 0.15. Only the neat FA (i.e., polymerized without any additional water) has a much lower DOS (i.e., 0.09).

Finally, insights on the nature of the carbonyls contained in the open structure of PFAs were given. The existence of esters in these open structured resins was highlighted, especially when FA polymerization was conducted without additional water. On the other hand, FA polymerization conducted with additional water rather highlighted the formation of ketonic groups in their opened structures and much fewer esters were identified.

To conclude, even a relatively low water content in FA (i.e., 30%) will enable to promote furan ring opening independently of the initiator employed. The absence of additional water on the other hand limits, but does not prevent, the formation of open structures that contain esters functions.

This investigation sheds new light on how side-reactions during FA polymerization can influence the final polymeric structure. The fact that FA polymerization is conducted under acidic catalysis in the presence of water (coming either from the condensation or added from the formulation) irrevocably leads to open structures containing ester and ketonic species. It might be interesting to investigate which already cross-linked PFA resins would be sensitive to ring opening occurring after complete polymerization.

## Data Availability

Data is contained within the article or Appendix A.

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
