# Peer review of "Conditions to Control Furan Ring Opening during Furfuryl Alcohol Polymerization"

_molecules, 2022, doi:10.3390/molecules27103212_

Round 1
Reviewer 1 Report
The manuscript "Conditions to control furan ring opening during furfuryl alcohol polymerization" is novel research work. Here authors have used several catalyst to see the polymerization from furfuryl alcohol to polyfurfuryl alcohol resin in presence and absence of water. I recommend for publication.
Author Response
We thanks the reviewer for his/her time spent on the manuscript and we are honored by his/her positive evaluation.
Reviewer 2 Report
The authors present a manuscript regarding the control of the ring opening during the poymerization of furfuryl alcohol. The introduction ans well as the conclusions are well written. The structure of the paper is nice and readable, I have only some minor points and I'd like to comment that the author Guigo self-cited 7 times up to 44 references: I think this is not ethic neither a correct behaviour.
Here the minor checks:
line 39,40,42,49: the references you have written in these line are in different format, please be homogeneous if not the result is not readable.
figure 1,2, etc: please either i the caption of the figure and on the axes, complete well the definitions: the figures have to be self-standing without reading the paper (i.e. resolve the acronym DOS)
Figure 4: as the residues R1 and R2 of the two molecules are not the same, please complete with other numbers.
Figure 4, line 288, etc: this simbol ~ means "same order of magnitude", please use the ≈
Author Response
Dear reviewer. We would like to thank you for your time spent on the manuscript and the positive comments you made about our article.
Regarding your concern about self-citations, we have attentively re-checked the 7 self-citations and we have suppressed 3 of them that are not dentrimental for the general comprehension. However, the remaining four self-citations are really important for understanding and discussing the current data. We are sharing your opinion regarding self-citations in the sense that an article should not only be made of self-citations.
Concerning your minor revisions:
1/line 39,40,42,49: the references you have written in these line are in different format, please be homogeneous if not the result is not readable.
Answer: many thanks. We have checked the format of all references to made them homogeneous.
2/figure 1,2, etc: please either i the caption of the figure and on the axes, complete well the definitions: the figures have to be self-standing without reading the paper (i.e. resolve the acronym DOS)
Answer: thanks for this suggestions. We have changed the names of the axes without DOS and rather Degree of open structures
3/Figure 4: as the residues R1 and R2 of the two molecules are not the same, please complete with other numbers.
Answer: Thanks for the suggestion. We have made one molecule with R1 and R2 and the other one with R3 and R4 (Figure 4)
4/Figure 4, line 288, etc: this simbol ~ means "same order of magnitude", please use the ≈
Answer: Indeed; many thanks. We have replaced the symbol ~ with the symbol ≈ both in Figure 4 and in the text.